# NAD^+^ Precursors: A Questionable Redundancy

**DOI:** 10.3390/metabo12070630

**Published:** 2022-07-09

**Authors:** Carles Canto

**Affiliations:** 1Nestlé Institute of Health Sciences, Nestlé Research Ltd., EPFL Campus, Innovation Park, Building G, 1015 Lausanne, Switzerland; carles.cantoalvarez@rd.nestle.com; Tel.: +41-(0)-216326116; 2School of Life Sciences, Ecole Polytechnique Fédérale de Lausanne (EPFL), 1015 Lausanne, Switzerland

**Keywords:** NAD^+^, vitamin B3, nicotinamide riboside, niacin

## Abstract

The last decade has seen a strong proliferation of therapeutic strategies for the treatment of metabolic and age-related diseases based on increasing cellular NAD^+^ bioavailability. Among them, the dietary supplementation with NAD^+^ precursors—classically known as vitamin B3—has received most of the attention. Multiple molecules can act as NAD^+^ precursors through independent biosynthetic routes. Interestingly, eukaryote organisms have conserved a remarkable ability to utilize all of these different molecules, even if some of them are scarcely found in nature. Here, we discuss the possibility that the conservation of all of these biosynthetic pathways through evolution occurred because the different NAD^+^ precursors might serve specialized purposes.

## 1. Introduction

Nicotinamide adenine dinucleotide (NAD^+^) is a vital molecule for most, if not all, forms of life. On the one hand, NAD^+^ is an important co-enzyme for hydride transfer enzymes that critically determine cellular metabolic pathways, including glycolysis, tricarboxylic acid (TCA), and mitochondrial oxidative phosphorylation. These dehydrogenase and oxidoreductase enzymes catalyze either the reduction (NAD^+^ to NADH) or oxidation (NADH to NAD^+^) of the NAD backbone [1].

NAD^+^ can also be phosphorylated by the NAD kinase enzyme (NADK) to generate NADP^+^—a structural analogue that incorporates a 2′-phosphate on the adenosine ribose moiety [1,2]. In this way, NADP^+^ and its reduced form, NADPH, constitute a second set of redox cofactors, although they are not exchangeable with NAD^+^/NADH. In this sense, NADP^+^ and NADPH are generally found at lower intracellular levels than NAD^+^ and NADH [2,3]. Moreover, while the balance between NAD^+^ and NADH is most often maintained favoring the reduced form, the opposite occurs with NADP^+^/NADPH [2,3]. NADP^+^/NADPH perform important functions in the handling of reactive oxygen species (ROS) and xenobiotic metabolism, as NADPH donates a hydride equivalent to reduced glutathione, reduced thioredoxin, and reduced peroxiredoxins [2]. NADPH also acts as an electron donor in the synthesis of fatty acid, steroid, and DNA molecules [2]. Finally, NADP^+^ is utilized in the pentose phosphate pathway to regenerate NADPH, which can then be used for nucleotide synthesis [2].

NAD^+^ is also used as degradation substrates for different families of enzymes, including ADP-ribosyl transferases (ARTDs, historically known as PARPs), sirtuins, and NAD glycohydrolases such as CD38, CD157, and SARM1 [3,4] (Figure 1). ARTDs break down NAD^+^ to mono- or poly-ADP-ribosylate target proteins, rendering nicotinamide (NAM) as a side product [3]. ARTD1 and ARTD2, which account for most ADP-ribosylation activity in the cell, are critical for DNA repair, and critically influence cellular metabolism by gauging NAD^+^ and ATP levels [5]. Sirtuins act as deacylase enzymes, removing short carbon-chain moieties from lysine residues in a wide range of proteins, including histones, transcriptional regulators, and metabolic enzymes [3]. Sirtuins cleave NAD^+^ to NAM while transferring the acyl group from the substrate to the ADP-ribose moiety of NAD^+^ [3]. Finally, NAD glycohydrolases (or NADases) generate cyclic ADP ribose (cADPr) from NAD^+^, also releasing NAM in the reaction [3,6]. The cADPr produced then acts as a second messenger for Ca^2+^ mobilization. NADases play critical roles in immune function and neurodegeneration [6].

## 2. Maintaining NAD^+^ Levels

The simple fact that multiple enzymatic activities break down NAD^+^ in eukaryote cells implies that we must have developed methods to synthesize NAD^+^ in order to maintain NAD^+^ levels. A second implication is that NAD^+^ consumption is not constant. Under some circumstances, such as the activation of ARTD1 (aka PARP-1) upon DNA damage, NAD^+^ is consumed at a higher rate and, therefore, NAD^+^ synthesis rates also need to increase to maintain NAD^+^ levels [5].

While variable depending on the cell type, tissue, and environmental conditions, NAD^+^ levels generally fluctuate in the range of 200–500 micromoles, according to enzymatic and mass spectrometry techniques ([3] and Table 1). NAD^+^-consuming enzymes have varying affinities towards NAD^+^. The Michaelis constant (Km) of an enzyme describes the substrate concentration at which half of the enzyme’s active sites are occupied by substrate. A high Km means that a lot of substrate must be present to saturate the activity of the enzyme (low affinity). Conversely, a low Km means that only a small amount of substrate is needed to maximize the activity of the enzyme (high affinity). Interestingly, given the Km of NAD^+^-consuming enzymes for NAD^+^, the activity of most of them should not be rate-limited by cellular NAD^+^ levels. However, these values should be cautiously interpreted. Firstly, measurements of intracellular NAD^+^ levels do not always distinguish between free and protein-bound NAD^+^, which might lead to an overestimation of the bioavailability of NAD^+^. Secondly, most of the measurements establishing these ranges also fail to reflect NAD^+^ levels at subcellular resolution, even when ample literature suggests that NAD^+^ metabolism can be regulated in a cellular-compartment-specific fashion. For example, NAD^+^ content is particularly higher in the mitochondrial compartment [3,7]. The recent development of fluorescent biosensors for NAD^+^ has allowed for new estimations of free cellular NAD^+^ content, revealing total intracellular values of ~106 µM, with a nuclear content of ~109 µM and ~230 µM in mitochondria [8]. These data confirm that a significant part of the NAD^+^ detected using classic enzymatic methodologies might be protein-bound. With these concentrations in mind, NAD^+^ content might be rate-limiting for the activity of some sirtuin enzymes under normal conditions. For example, the Km of SIRT1—the most well-studied sirtuin—for NAD^+^ is ~95 micromoles, which is very close to the estimated nuclear NAD^+^ content [7,8]. Therefore, nuclear SIRT1 might be responsive to alterations in the availability of NAD^+^. In line with this, increased NAD^+^ consumption by ARTD1—another nuclear enzyme—generally leads to lower SIRT1 activity [9]. A similar case could be made for SIRT3 or SIRT5, which are mitochondrial sirtuins with a very high Km for NAD^+^ [7]. In contrast, ARTD1 and CD38 show a very low Km for NAD^+^, suggesting that their activity is not generally bottlenecked by NAD^+^ content [7], unless a very sharp decline in NAD^+^ occurs.

NAD^+^ has a relatively fast turnover at the whole- body level, albeit it can dramatically differ across mammalian tissues. Labelling studies demonstrated that mouse livers display a NAD^+^ turnover of around ~2 h, which largely differs from skeletal muscle, where the turnover is ~8–10 times slower [10]. This constant requirement for NAD^+^ synthesis can be met by using 2 independent routes. First, through the internal regeneration of NAD^+^ stores by using an endogenous metabolite as a precursor. Second, through the dietary intake of molecules that can serve as NAD^+^ precursors. These dietary molecules that serve as dietary precursors are globally termed Vitamin B3s [1]. In the past, they were also known as niacin, even if nowadays niacin is mostly used to refer to a particular NAD^+^ precursor molecule, nicotinic acid (NA). Human studies suggest that the dietary intake of 20 mg of niacin (or niacin equivalents) is enough to meet daily requirements [1]. Such as small need for NAD^+^ precursors suggest that mammalian organisms mostly use internal recycling systems (salvage routes) to maintain tissue NAD^+^ levels.

While small, the requirement for dietary NAD^+^ precursors is critical. This is clearly illustrated by the fact that a deficiency in dietary vitamin B3 leads to the development of pellagra—a medical condition characterized by diarrhea, dermatitis, and dementia [1]. If untreated, pellagra is fatal. However, pellagra is the dramatic manifestation of a very severe dietary deficiency of NAD^+^ precursors. In recent years, several lines of evidence have highlighted how more modest decreases in NAD^+^ levels can predispose mammalian organisms to the development of metabolic complications. For example, decreased NAD^+^ content has been observed in the livers of mice and human patients suffering from hepatic damage [11,19,20,21]. Similarly, NAD^+^ levels decline with age in skeletal muscle, in great part due to physical inactivity [22]. Even if lower NAD^+^ levels might not necessarily be causal for diet- or age-related complications, supplementation with NAD^+^ precursors has been consistently shown to slow down or alleviate the progression of multiple metabolic and neurodegenerative disorders (for dedicated reviews, please see [23,24,25]). Therefore, the maintenance of NAD^+^ levels plays a critical role in health and disease.

### 2.1. Dietary NAD^+^ Precursors

Several dietary molecules can act as NAD^+^ precursors. In most animal products and uncooked foods, their intracellular NAD^+^ and NADP^+^ content accounts for most of our provision of dietary NAD^+^. However, this is not achieved through the direct absorption of NAD^+^ or NADP^+^, as these molecules cannot passively cross cellular membranes. Classic studies using radiolabeled compounds in perfused rat intestines demonstrated that most of the ingested NAD^+^ is hydrolyzed in the small intestine [26]. Initially, NAD^+^ is transformed into nicotinamide mononucleotide (NMN) and 5′-AMP by pyrophosphatase enzymes. NMN is subsequently hydrolyzed to nicotinamide riboside (NR), which is then slowly converted to NAM [26,27]. Alternatively, NAM can also be generated by the direct cleavage of NAD^+^, generating diverse ADP-ribose metabolites [26]. In these experiments, NAM was poorly deamidated to NA in the gut [27] (Figure 2).

However, recent experiments in living mice—either germ-free or treated with antibiotics—indicate that a substantial part of orally administered NR and NAM is metabolized to NA by the gut microbiome (Figure 2). This was originally hinted at by the serendipitous finding that oral supplementation with NR led to increased circulation levels of nicotinic acid adenine dinucleotide (NAAD) [12]—a deamidated metabolite from the Preiss–Handler pathway that is not an intermediate of the direct conversion path from NR to NAD^+^ (Figure 3). A second study using isotope-labelled tracer molecules demonstrated that 3 h after the oral administration of NR or NAM, most of their incorporation in hepatic NAD^+^ occurred via their prior conversion to NA [28]. This was confirmed in a third study, which also examined earlier time points. Interestingly, this study reported how orally administered NR and NAM increased NAD^+^ levels in a biphasic manner [29]. In an early phase (within 1 h after administration), NR and NAM were directly taken up by the small intestine and utilized for NAD^+^ synthesis in the liver [29]. In a later phase (3 h after administration), the transformation of both compounds to NA by the gut microbiome was the main driver of the sustained increase in NAD^+^ synthesis. Therefore, an integrative view of these studies (Figure 2) suggests that dietary NAD^+^ is broken down in our intestines to NMN and NR. NR can be partially absorbed as such in an early phase of digestion, but it is slowly degraded to NAM either in the gut lumen, by intestinal cells, or once placed in circulation [10,13,26,29]. NAM produced in the intestinal lumen can also reach circulation, but the gut microbiome slowly converts NAM into NA, which is the substrate for the sustained hepatic NAD^+^ production after a meal [28,29]. The time and effectiveness of NAM’s conversion to NA might depend on the gut microbiome composition, which can vary across species and individuals.

Recent studies have also analyzed how direct intravenous (IV) administration of NAD^+^ can influence the NAD^+^ metabolome. In humans, the infusion of NAD^+^ confirmed that NAD^+^ must be quickly metabolized, and only 2 h after the initiation of IV infusion, products of NAD^+^ metabolism could be detected in plasma and urine [18]. Unfortunately, the study did not explore intracellular NAD^+^ metabolism in circulating cells, but the results clearly suggest that NAD^+^ is not a stable circulating molecule and that, as in the gut, it is quickly degraded to other molecules that can be used as intracellular precursors.

### 2.2. Cellular Uptake to Intracellular Metabolism

Shockingly, NA and NAM are found in circulation at rather low concentrations. NA has been reported at 80–180 nM concentrations in human plasma [18,30], while NAM can be found in the very low micromolar range [18,30,31,32]. Interestingly, these concentrations are insufficient to increase NAD^+^ levels in cultured cells. The reason why cells can generally sustain their NAD^+^ levels and, therefore, their viability, is because they can salvage NAD^+^ via multiple endogenous and exogenous precursors.

NA can enter cells via SLC transporters, including SCL22A13 [33] or SCL22A7 [34], amongst others. These transporters can act as organic anion exchangers, and the influx of NA is maintained by its quick intracellular conversion to nicotinic acid mononucleotide (NAMN) by the nicotinic acid phosphoribosyltransferase (NAPRT) enzyme. NAMN is then adenylated by the nicotinamide mononucleotide adenylyl transferases (NMNATs), generating nicotinic acid adenine dinucleotide (NAAD), which is finally amidated by the NAD^+^ synthase (NADSYN) to produce NAD^+^ [3] (Figure 3). This route of NAD^+^ synthesis is generally known as the Preiss–Handler pathway. Interestingly, tryptophan (Trp) can also be used as a substrate for NAD^+^ synthesis through the de novo pathway (Figure 1). The de novo pathway leads to the synthesis of NAMN, converging with the Preiss–Handler pathway [3]. However, Trp is a very weak NAD^+^ precursor, especially in humans, as it is largely routed towards other cellular functions [35].

Understanding cellular NAM utilization has proven to be challenging, as NAM might be able to diffuse through cellular membranes, and no known transporter exists [3]. Once inside the cell, NAM can be transformed to nicotinamide mononucleotide (NMN) by the nicotinamide phosphoribosyltransferase (NAMPT) enzyme [36]. Then, NMN can be adenylated by NMNATs to form NAD^+^ [3,36] (Figure 3). As mentioned in previous sections, NAM is a common side product of the reactions catalyzed by NAD^+^-consuming enzymes. Therefore, using this NAM to synthesize NAD^+^ has a twofold benefit: First, it allows cells to recycle intracellularly produced NAM and rely less on exogenous precursors. This could have been evolutionarily important in situations of food scarcity, and explains why mammalian organisms largely rely on this pathway to sustain NAD^+^ levels in different tissues [37,38]. Second, it prevents NAM accumulation, which could exert end-product inhibition of NAD^+^-consuming enzymes. In this sense, mathematical models revealed that a second enzymatic activity might have been implemented in vertebrate organisms in order to prevent NAM accumulation [39]. This activity is the nicotinamide *N*-methyltransferase enzyme (NNMT), which catalyzes the methylation of NAM into methyl-NAM (mNAM)—a metabolite that is generally diverged for excretion (Figure 2).

NR can also be used by eukaryote cells to synthesize NAD^+^ [40]. In mammalian cells, NR is directly transported into the cells via equilibrative nucleoside transporters (ENTs) [13,41]. Its first intracellular step towards NAD^+^ biosynthesis is its phosphorylation by nicotinamide riboside kinases (NRKs), forming NMN, which is then transformed to NAD^+^ by NMNATs [13,40] (Figure 3). Depending on the cell type, NR can also lead to NAD^+^ synthesis through the cleavage of its NAM ring by the purine nucleoside phosphorylase enzyme (PNP1) [42]. Given that NMN is an intermediate metabolite of NR- and NAM-induced NAD^+^ synthesis, multiple works have experimented with the possibility of directly administering NMN as an NAD^+^ precursor. While exogenous NMN can lead to NAD^+^ synthesis, there is some controversy as to how this occurs. Genetic and pharmacological approaches initially demonstrated that NMN, like other nucleotides, fails to cross the plasma membrane as such. Instead, it must be extracellularly dephosphorylated to NR, which is then transported into the cell [13,41,43]. However, a recent report proposes that SLC12A8 could act as an NMN transporter [44]. This finding has raised some controversy [45], and future research will be needed to elucidate physiological scenarios in which the direct transport of NMN occurs, and whether this plays a significant role in NAD^+^ maintenance. A final precursor to consider in relation to the NRK pathway is the deamidated form of NR—nicotinic acid riboside (NAR). Through its phosphorylation by NRKs, NAR is converted to NAMN, which then feeds the classic Preiss–Handler pathway [3]. Accordingly, extracellular NAR can serve as an NAD^+^ precursor in eukaryote cells [46,47].

More recently, another pathway came into play with the discovery of dihydronicotinamide riboside (NRH) as a new NAD^+^ precursor [15,48]. NRH is structurally similar to NR, but with a reduction in the NAM ring (Figure 3). NRH is also transported into the cells through ENTs [15]. To generate NAD^+^, NRH needs to be phosphorylated into dihydronicotinamide mononucleotide (NMNH)—a reaction catalyzed by the adenosine kinase (AK) enzyme [15,49] (Figure 3). NMNH is later transformed into NADH by NMNAT enzymes [15], and NADH is finally oxidized to NAD^+^. As with NMN, exogenous NMNH can also act as an NAD^+^ precursor in mammalian cells, but it requires prior dephosphorylation to NRH to be transported into the cell [50].

## 3. NAD^+^ Precursors: More of the Same?

The fact that eukaryote cells have exquisitely conserved so many different pathways to generate NAD^+^ leads to the question of whether all of these different precursors are functionally redundant. Circulating precursors are indispensable, yet they are most often found in plasma at concentrations that are unlikely to be able to sustain NAD^+^ levels, suggesting that once in circulation, they are very rapidly taken up by cells and tissues. Supporting this latter point, tracer analyses have revealed how NR and NMN disappear from the bloodstream and engage NAD^+^ synthesis within minutes after intravenous delivery [10]. Hence, physiologically relevant fluctuations in the circulatory levels of these precursors might exist, yet could be difficult to detect. Some of these fluctuations might respond to dietary intake, but the uptake and release of cellular NAD^+^ precursors might show additional layers of complexity, as is discussed in this section.

### 3.1. Uniqueness in NAD^+^ Precursors: Actions beyond NAD^+^ Metabolism

NA has been used in the clinic for more than 50 years as a lipid-lowering drug [51]. A marked secondary effect of NA treatment is that it prompts a severe flushing reaction in patients. This flushing does not derive from the action of NA as an NAD^+^ precursor, but rather from the activation of a G-coupled receptor—GPR109A—in Langerhans cells [52]. Given the relatively low levels of NA in the blood, the activation of this receptor is unlikely to be a native function of NA, but rather an effect of pharmacological dosing. Several lines of evidence suggest that the beneficial effects of NA on plasma lipids are also mediated via GPR109A’s activation, rather than through its effects as an NAD^+^ precursor [53]. Accordingly, the clinical administration of alternative NAD^+^ precursors, such as NAM or NR, does not significantly alter blood lipid profiles [16,53,54]. However, this is not an entirely clear picture. While NA leads to decreased LDL and increased HDL levels in circulation, GPR109A is poorly expressed in the liver, which is the main tissue controlling whole-body lipoprotein metabolism [53,55,56]. It was argued that the beneficial effects of NA on blood lipid profiles derive not from hepatic effects, but from the activation of GRP109A in adipose tissue, which leads to an anti-lipolytic effect [57]. This, in turn, would prevent adipose-tissue-derived free fatty acids from being re-esterified in the liver and secreted as VLDL particles. However, works with mice lacking GPR109A and with GPR109A agonists have challenged this hypothesis. When GPR109A-deficient mice were treated with NA, the acute anti-lipolytic effects of NA were prevented, yet this did not influence the benefits of NA on blood lipid profiles, such as increasing HDL cholesterol while reducing LDL cholesterol [58]. Similarly, GPR109A agonists failed to improve blood lipid profiles [58]. Therefore, while NA blocks lipolysis through GPR109A, this does not explain the effects of NA on blood lipid profiles.

Other NAD^+^ precursors, such as NAM, NR, or NMN, do not activate GPR109A in cultured cells [55,59]. This is consistent with mutagenesis efforts indicating that NA-related activation of GPR109A relies on the interaction between the carboxylic acid group of NA and Arg111 in one of the transmembrane regions of the receptor [60]. It would be interesting to know whether other NA-related molecules, such as NAR or the methylated form of NA—trigonelline—can also act as GPR109A agonists.

As mentioned in Section 2.1, recent studies in murine models suggest that a very substantial amount of orally administered NAM, NR, or NMN is transformed to NA by the gut microbiome [28,29]. This could explain why large increases in NAAD are observed in circulation and in tissues after the administration of NR, whether in mice or in humans [12,16,17]. However, given the very high affinity of GPR109A for NA, and the fact that NA-induced flushing has been reported at oral concentrations > 50 mg per day [51,52,55], one would expect that a high rate of conversion of NAM or NR to NA in the gut would also manifest in flushing. Strikingly, no significant flushing side effects have been reported with NR, even at gram dosages [16,54]. This could suggest that in humans the conversion of NR/NAM to NA in the gut lumen might occur at a slower rate—avoiding a high increase in NA in circulation—or that NA is quickly metabolized into other metabolites.

The example of NA illustrates that NAD^+^-metabolites might be able to exert biological actions by interacting with receptors or proteins unrelated to NAD^+^ metabolism. Thus, the therapeutic and clinical properties of NA are clearly different from those of other NAD^+^ precursors.

### 3.2. Uniqueness in NAD^+^ Precursors: Enzymatic Control on NAD^+^ Consuming Enzymes

If the effects of NA on chronic blood lipid reduction were NAD^+^-related, one has to wonder why NAM failed to promote them. One of the reasons that could explain this circumstance is that NAM, unlike other NAD^+^ precursors, is also the product of the enzymatic activities of sirtuins, ARTDs, and NADases. As such, NAM can exert end-product inhibition of these enzymes. Initial evidence for this was obtained by Bitterman et al., who demonstrated that NAM could inhibit yeast Sir2 and human SIRT1 noncompetitively in vitro [61]. The inhibitory constant for NAM in various sirtuin enzymes has been determined to be in the range of 30–200 μM. NAM is present in human plasma at around 10 μM levels [17]. Intracellular concentrations have been reported at ~100 μM in murine embryonic stem cells [62], while in skeletal muscle they were estimated at ~80 μM [17]. This suggests that intracellular fluctuations of NAM could influence sirtuin activity. Similarly, most pharmacologically developed ARTD inhibitors are NAM mimetics that bind in the NAD^+^-binding pocket of the catalytic domain [63]. With this in mind, it is not surprising that NAM is commonly used to blunt sirtuin or ARTD activities, either in cultured cell assays or in lysis buffers. For this, however, NAM is used at millimolar concentrations, which are rarely attained under physiological conditions, as NAMPT and NNMT activities prevent NAM accumulation. This latter factor could also explain why NAM concentrations are rarely affected in murine or human tissues. An intraperitoneal injection of NAM, NR, or NMN (500 mg/kg) prompted a large increase in circulating NAM levels after an hour, yet did not significantly alter intrahepatic NAM levels [13]. Similarly, chronic NR administration in humans increases NAM levels in circulation, yet not in skeletal muscle [17]. Together, these observations suggest that even if NAM can act as an inhibitor of NAD^+^-consuming enzymes in vitro, the outcomes of in vivo NAM administration are more complex. In vivo, low doses of NAM might foster NAD^+^ and mNAM synthesis, facilitating the activity of NAD^+^-consuming enzymes. However, at high doses, NAM levels might surpass the enzymatic capacities of NAMPT and NNMT, leading to NAM accumulation and the inhibition of sirtuins or ARTDs. This could explain why the literature has shown effects of NAM administration consistent with both activation and inhibition of sirtuin [64]. Furthermore, it might explain the hepatotoxic effects of NAM supplementation at high doses—generally >3 g/day in humans [65]. From an evolutionary perspective, it makes sense that organisms have developed ways to protect against the accumulation of a metabolite that might diffuse across membranes and could inhibit critical enzymes for cellular function. However, this opens several questions: What is the NAM dosage required to prompt accumulation? Does this threshold differ across tissues or cell types? Is this threshold affected in pathophysiological situations?

Several elements can influence NAM levels in the cells. On the one hand, NAM levels might increase as a result of increased NAD^+^ degradation through NAD^+^-consuming enzymes. This could be the case for aged and obese mice, where increased ARTD activity has been observed [9,66]. In line with this, isotope tracing combined with mass spectrometry analyses identified higher NAD^+^ consumption rates in tissues from aged mice [67]. A second element influencing NAM levels could be alterations in NAMPT and NNMT activities. In this sense, NAMPT levels decrease with age and with metabolic disease in several murine tissues, including adipose tissues and the liver [11]. In contrast, NNMT levels are increased in the liver and adipose tissue of obese and diabetic mice [14,68]. Consequently, obesity might lead to increased production of NAM, the use of which as an NAD^+^ precursor via NAMPT might be compromised, leading to NAM accumulation unless it is shunted towards methylation via the NNMT pathway. Such scenarios significantly complicate our ability to forecast an unequivocal consequence of NAM supplementation in terms of its impact on NAD^+^ levels and the activity of NAD^+^-consuming enzymes.

Recently, another molecule that has received attention due to its direct influence on the activity of NAD^+^-consuming enzymes is NMN. This concept emerged from the observation that the overexpression of NMNAT enzymes could mediate robust protection against axotomy-induced axonal degeneration. While this was initially thought to be the consequence of increased NAD^+^ synthesis capacity, NAD^+^ levels were similar in neurons overexpressing NMNAT [69,70]. Even more puzzling was the fact that FK866—an NAMPT inhibitor—could also acutely prevent axonal degeneration [70,71]. The above observations raised the possibility that the benefits of NMNAT stemmed from preventing NMN accumulation, rather than increasing NAD^+^. Accordingly, administration of NMN at concentrations between 25 and 75 μM was enough to prompt axonal degeneration in FK866-treated neurons [71]. However, the mechanism by which NMN could prompt such effects is not known.

A major breakthrough in the field occurred when the SARM1 protein was identified as a critical mediator of axon death [72]. SARM1 is an enzyme that acts as an NADase, and this NADase activity is required for axonal degeneration [73]. NMN, but not NR or NAMN, allosterically activated SARM1, with a half-maximal concentration in the low micromolar range [74,75]. This generated a model in which a decrease in NMNAT2 expression after axonal damage [76] would facilitate the accumulation of NMN, leading to SARM1 activation and a sharp increase in the consumption of NAD^+^, which ultimately elicited axonal death. Hence, the direct binding of NMN to SARM1 constitutes a perfect example of how specific NAD^+^-related metabolites can drive enzymatic functions irrespective of their role as NAD^+^ precursors. This is important when considering supplementation strategies based on cell-permeable NMN analogs, as they could accumulate enough to trigger SARM1 activity [75].

Interestingly, recent works have highlighted how SARM1 might be regulated by multiple NAD^+^ metabolites. Figley et al. demonstrated how SARM1 is not controlled simply by NMN, but actually by the ratio between NMN and NAD^+^ levels, gauging NMNAT activity in this way [77]. Furthermore, NAMN has been shown to inhibit SARM1 by competing with the binding of NMN [78].

### 3.3. Uniqueness of NAD^+^ Precursors: Tissue Expression

An important element determining the action of NAD^+^ precursors is the expression of the enzymatic machinery required to transform them into NAD^+^. The action of NAD^+^ precursors does not seem to be rate-limited by their transport into the cells, which generally occurs through rather unspecific transporters [23]. The main exception is NAM, which has no identified transporter, and whose rate of diffusion across cellular membranes is not completely understood [23]. In most cell types tested to date, NAM-induced NAD^+^ synthesis is rate-limited by the phosphoribosyltransferase step catalyzed by NAMPT [36]. In mice, NAMPT protein levels are high in brown adipose tissue (BAT), along with the heart, muscle, liver, and kidneys, but weak in other tissues, such as the brain, pancreas, spleen, or white adipose tissue (WAT) [79]. However, even in organs and tissues with low NAMPT expression, NAMPT levels can still be notably high in specific cell types. This is the case for the pancreas, where NAMPT is enriched in β-cells [80], and the brain, where NAMPT is enriched in neurons [81].

NAMPT can also be cleaved and secreted as an extracellular enzyme (eNAMPT). While eNAMPT was initially thought to facilitate the conversion of NAM into NMN in the bloodstream, this view was challenged by the very low levels of the enzymatic substrates or products of this reaction in circulation [82]. More recently, eNAMPT has been shown to be carried through systemic circulation in extracellular vesicles, in both mice and humans [83]. Then, eNAMPT vesicles can be internalized by other cells and tissues to enhance NAD^+^ biosynthesis [83].

NAMPT allows recycling of the NAM produced through the activities of NAD^+^-consuming enzymes into NAD^+^. The work on tissue-specific NAMPT-knockout murine models has shown that, in fact, NAM salvage is critical to maintain baseline NAD^+^ levels, albeit to varying degrees in different cells and tissues. For example, deletion or pharmacological inhibition of NAMPT promotes a very sharp decline in muscle, but a more modest reduction in the liver [37,38,84].

The above observation raises the question of why hepatic NAD^+^ levels are more resilient to NAMPT deletion than those in muscle. This is shocking, considering that NAD^+^ turnover in liver is largely higher than in muscle [10]. The answer might be related to the fact that the liver also expresses the molecular machinery to use multiple other NAD^+^ precursors. For example, the liver expresses NAPRT—the rate-limiting enzyme in the Preiss–Handler pathway. The NAPRT protein is abundant in the liver and kidneys, yet absent in muscle [85]. Another unique enzyme of the Preiss–Handler pathway is NADSYN, which is also expressed in the liver and kidneys, but not in muscle [86]. This suggests that NA can act as an NAD^+^ precursor in some tissues (e.g., the liver and kidneys), but not in others (e.g., muscle). Supporting this, the intraperitoneal injection of NA failed to increase NAD^+^ in muscle, while it only needed 5 min to prompt a large increase in hepatic NAD^+^ content [87]. These observations also suggest that other deamidated NAD^+^ precursors, such as NAR, are only effective in tissues harboring molecular machinery of the Preiss–Handler pathway.

In the case of NR, its action as an NAD^+^ precursor is rate-limited by NRK activity. The NRK1 protein can be detected in most tissues, yet its highest levels are found in the liver and kidneys [13]. Accordingly, NR has greater effects than NAM on liver and kidney NAD^+^ content 1 h after intraperitoneal injection (500 mg/kg). Surprisingly, despite the relatively weak expression of NRK1 in muscle, NR had slightly greater effects on NAD^+^ levels than NAM [13]. This might be explained by the existence of NRK2 in muscle [13,88]. Interestingly the expression of the *Nmrk2* gene, encoding the NRK2 protein, seems to be highly responsive to stress conditions. This was initially shown by the Milbrandt lab, who reported a large increase in *Nmrk2* mRNA levels in dorsal root ganglia after sciatic nerve transection [89]. Similarly, NRK2 protein and mRNA levels largely increased in the failing heart, in both mice and humans [90]. The increase in NRK2 occurred in parallel to a decline in NAMPT levels [90]. This led to the hypothesis that NRK2 increases in order to optimize NR-driven NAD^+^ synthesis in situations where the use of NAM might be compromised. In Section 3.4, we further explore why these situations might occur.

### 3.4. Uniqueness of NAD^+^ Precursors: Balancing Metabolic Routes

Under normal diets, mice lacking enzymes for the Preiss–Handler pathway—either NAPRT or NADSYN—did not show alterations in hepatic NAD^+^ levels [29]. Similarly, healthy, young, NRK1-deficient mice showed no alterations in NAD^+^ levels across different tissues [13]. These findings are consistent with the concept that NAM might be the greatest contributor to the basal maintenance of NAD^+^ levels in most mammalian tissues. Accordingly, the specific deletion of NAMPT in the liver using classic Cre/Lox strategies led to a reduction in hepatic NAD^+^ content, even in young healthy mice [38]. This is also consistent with classic reports suggesting that NAM is a commonly absorbed NAD^+^ precursor [26,27,87]. However, most recent works using elegant tracer analyses indicate that NAM in the digestive tract is largely deamidated to NA in the gut lumen [28,29]. With that being the case, why do NAPRT- or NADSYN-knockout mice not display defects in hepatic NAD^+^ levels? Recently, Yaku et al. proposed that deamidated-to-amidated base changes can occur in these circumstances through the enzymatic activity of BST1 (also known as CD157), hence allowing cells to generate NAM from NA and sustain NAD^+^ synthesis when the Preiss–Handler pathway is not effective [29]. Similarly, tracer experiments also revealed that tissues can better use NR as a precursor when NAM or NA salvage pathways are impeded [29,37].

NRK1-KO mice do not show defects in NA- or NAM-induced NAD^+^ synthesis [13]. However, when liver-specific NRK1-KO mice are placed on a high-fat diet (HFD), they display a deficiency in hepatic NAD^+^ levels [14]. This occurred in parallel to mitochondrial dysfunction and hepatic insulin resistance [14]. Surprisingly, the dietary supplementation with NAM was unable to rescue NAD^+^ levels or the physiological deficiencies of the NRK1-deficient mice [14]. This work opened the possibility that, in some physiological scenarios, NR could have a unique function. Supporting this, other works have also observed how the effects of NR supplementation cannot always be phenocopied by NAM or NA [91,92]. The uniqueness of NR might rely on the fact that it is a nucleoside, and does not require the initial incorporation of ribose into the molecule in its build-up towards NAD^+^ synthesis. In contrast to NR, NA and NAM require a ribose donor—phosphoribosyl pyrophosphate (PRPP)—in the steps catalyzed by NAPRT and NAMPT (Figure 3). Most PRPP is generated by PRPP synthetases, which transform ribose 5-phosphate and ATP into PRPP and AMP [93]. However, PRPP is not exclusively used for NAD^+^ biosynthesis. Instead, it constitutes an essential cofactor for the biosynthesis of purine and pyrimidine nucleotides, as well as the amino acids histidine and tryptophan [93]. Therefore, NAD^+^ biosynthesis might compete for PRPP’s availability with multiple other metabolic pathways. During high-fat feeding, hepatic PRPP levels decline [94], which could be a consequence of the fact that the liver undergoes higher DNA damage, apoptosis, and compensatory proliferation, resulting in higher nucleotide synthesis rates [14]. A decline in PRPP could bottleneck the ability of the liver to generate NAD^+^ from NA or NAM. The use of NR as an NAD^+^ precursor can bypass this limitation. This could explain why NRK1-KO mice cannot sustain hepatic NAD^+^ levels when placed on an HFD. PRPP deficiencies have also been reported in the liver and kidneys in situations of diabetes and starvation [94,95]. One could speculate that the need to ensure NAD^+^ synthesis during starvation, when PRPP levels might be limiting, could explain the exquisite conservation of pathways devoted to the use of ribosylated NAD^+^ precursors.

The need to preserve PRPP levels may also explain why NAM largely diverged towards mNAM synthesis in HFD-fed mice [14]. This could be prompted by the increase in NNMT expression in the livers of HFD-fed mice [14]. The conversion of NAM into mNAM may have some additional metabolic consequences. First, it has been proposed that high NNMT activity could reduce the availability of the universal methyl donor S-adenosyl methionine (SAM), potentially compromising other cellular needs for one-carbon metabolism, including epigenetic methylation events [96]. While mNAM is generally considered to be a metabolite targeted for excretion, it can also have cellular effects. Most notably, several reports have highlighted how mNAM can increase H_2_O_2_ production, either through Complex I inhibition [97], or by acting as a substrate for aldehyde oxidase enzymes [98]. Therefore, when using NAD^+^ precursors for therapeutic purposes, one must consider whether limiting PRPP levels or excessive diversion towards mNAM production can influence the outcomes of the intervention. In this sense, while high doses of NR did not lead to improvements in insulin sensitivity in pre-diabetic patients [16,54], low doses of NMN improved insulin sensitivity in pre-diabetic women [99]. Provocatively, one could hypothesize that a lower dose of NMN/NR might be enough to achieve metabolic benefits while preventing a surge in mNAM, which could lead to secondary effects neutralizing this benefit.

### 3.5. Uniqueness of NAD^+^ Precursors: Bioavailability

Surprisingly, no studies have compared the pharmacokinetics, pharmacodynamics, and biological effects of the different known NAD^+^ precursors side by side. An attempt to achieve this goal was performed by Trammell et al., who compared how oral administration of NA, NAM, and NR elevated the NAD^+^ metabolome in mice [12]. The results illustrated that the three compounds altered the hepatic NAD^+^ metabolome, with different dynamics. NA produced the lowest increase in hepatic NAD^+^, but its maximal action was faster than that of NR and NAM. More precisely, hepatic NA and NAD^+^ levels peaked at 15 min and 2 h after NA administration, respectively [12]. NAM gavage did not substantially increase hepatic NA at any point, and drove increased hepatic NAD^+^ accumulation from 2 to 8 h, peaking at the latter time. These observations are very consistent with previous reports using radioactive tracers [87]. NR elevated hepatic NAD^+^ by more than fourfold, peaking at 6 h post-gavage [12], suggesting that oral NR has different hepatic pharmacokinetics than oral NAM and NA. These observations are slightly at odds with more recent reports suggesting that NR and NAM are largely converted to NA in the gut lumen [28,29]. If this was the case, one should expect rather similar outcomes, unless the conversion of NR or NAM to NA by the microbiome is a very slow and/or partial process.

Another element to consider when using NR as a therapeutic agent is that it might degrade in circulation. Accordingly, when NR is intraperitoneally injected, orally gavaged, or incubated in isolated murine plasma, a large increase in NAM is observed [12,13]. The enzymatic activity responsible for the cleavage of NR into NAM in plasma is not known, although several works suggest that it could be the purine nucleoside phosphorylase 1 (PNP1) enzyme [42,100,101]. In line with these points, intact NR only partially reaches tissues after oral gavage [10,29]. Time-course experiments revealed that in the initial phase after oral gavage (within the first hour), NR is directly absorbed, and promotes hepatic NAD^+^ synthesis [29]. This might explain why some of the alterations in liver- and muscle-specific NAMPT-KO models can be rescued by NR [37,84,102,103]. Therefore, and despite being largely converted to NAM or NA, enough NR can reach tissues to promote biological effects. Tracer analyses on the muscle-specific NAMPT-KO mice also suggest that more intact NR reaches the muscle tissue in this model than in regular control mice [37]. This could be related to the increases in NRK2 often observed in response to tissue damage, as discussed in Section 3.3.

Interestingly, the reduction of NR in the NAM ring is enough to prevent its degradation in plasma [15]. This reduced form of NR, also known as NRH, can be detected in circulation at micromolar ranges after oral administration (250 mg/kg) [15]. The conversion of NRH into NAD^+^ is initiated by AK [15,49], which is ubiquitously found across mammalian organs [104]. Therefore, NRH is predicted to act as an NAD^+^ precursor in most tissues. However, the bioavailability and pharmacokinetics of NRH still need to be determined. In a fascinating twist, our work detected how the administration of NRH in mice led to detectable amounts of NR in circulation after 1 h [15]. Strikingly, we could not detect circulating NR after 1 h when NR was administered directly [15]. This suggests that NR is actively produced/released by murine tissues after NRH administration. In line with this, intracellular NR levels were dramatically increased in hepatocytes treated with NRH, and NR production required the transformation of NRH into NAD^+^ [15]. Therefore, under the right circumstances, NR might be synthesized and released as a circulating NAD^+^ precursor. The fact that cells might be able to synthesize NR from other NAD^+^-related molecules could also explain why NR can be found at micromolar concentrations in mammalian milk [40,105].

## 4. Conclusions

Critical developments in the past decade—such as the use of murine models deficient in specific NAD^+^ biosynthetic pathways, the establishment of metabolomic techniques, and the chemical generation of multiple isotope-labelled tracers—have provided a solid basis to hypothesize that NAD^+^ precursors are not entirely redundant.

Different cells and tissues make different uses of NAD^+^ precursors to sustain their NAD^+^ pools, as well as to generate other metabolites. Furthermore, the gut microbiome seems to be a critical determinant of the amounts and types of dietary NAD^+^ precursors that are finally incorporated in circulation. This has important implications for the use of NAD^+^ precursors as therapeutic agents. For example, the lack of NAPRT expression in muscle compromises the ability of NA to have any direct effect on its NAD^+^ content. Conversely, tissues with a high NAD^+^ turnover might be more susceptible to NAD^+^-boosting strategies. It must also be taken into consideration that pathophysiological situations influence the ways in which tissues might use different NAD^+^ precursors. This might occur through changes in the levels of enzymes or coenzymes participating in NAD^+^ metabolism.

Multiple lines of evidence also highlight how we might miss significant amounts of information when using NAD^+^ as the only readout of the actions of NAD^+^ precursors. NAD^+^ metabolic pathways are heavily intertwined, and do not have NAD^+^ as the sole and ultimate endpoint. For example, recent analyses suggest that dosing NA and NAM can lead to the generation and release of NAR and NR, respectively [106]. Similarly, NAM can lead to NAD^+^ biosynthesis, but also to that of mNAM and other downstream metabolites, some of which can have biological actions of their own.

Collectively, these observations indicate that we are only scratching the surface in our understanding of NAD^+^ metabolism. The field has sustained an incredible evolution during the last decade, even unveiling new NAD^+^ biosynthetic pathways. One would think that a century after their initial description, our understanding of NAD^+^ precursors would be very complete. Rather, we are just at the beginning of a riveting adventure.

## Figures and Tables

**Figure 1 metabolites-12-00630-f001:**
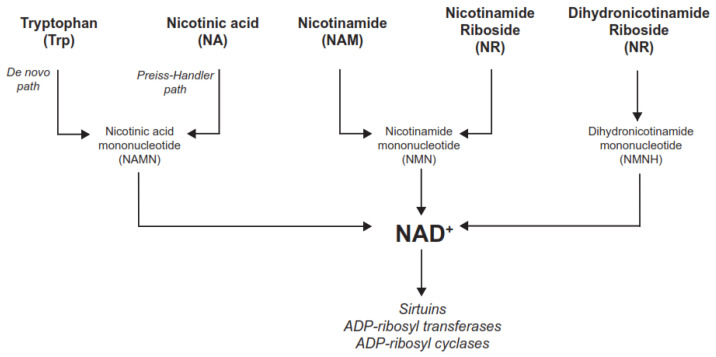
General scheme of NAD^+^ synthesis and degradation: NAD^+^ synthesis can be engaged from multiple precursors (in bold). Tryptophan (Trp) uses a path involving multiple reaction steps (de novo path). Nicotinic acid (NA), often referred to as niacin, also serves as an NAD^+^ precursor. Trp and NA lead to the synthesis of a common intermediate mononucleotide molecule—the nicotinic acid mononucleotide (NAMN), which is further transformed into NAD^+^. Nicotinamide (NAM) can also be used to prime NAD^+^ synthesis. Similarly, nicotinamide (NAM) and nicotinamide riboside (NR) can also be used as NAD^+^ precursors, converging into the synthesis of nicotinamide mononucleotide (NMN), which is a common intermediate for NAD^+^ synthesis. Finally, dihydronicotinamide riboside (NRH) is a recently discovered precursor using yet another independent path, leading to the generation of dihydronicotinamide mononucleotide (NMNH) and, ultimately, NAD^+^. In the figure, NRH is depicted in its 1-4 form, but 1-2 and 1-6 forms might exist. NAD^+^ can then be used by different families of enzymes, such as sirtuins, ADP-ribosyl transferases, or ADP-ribosyl cyclases.

**Figure 2 metabolites-12-00630-f002:**
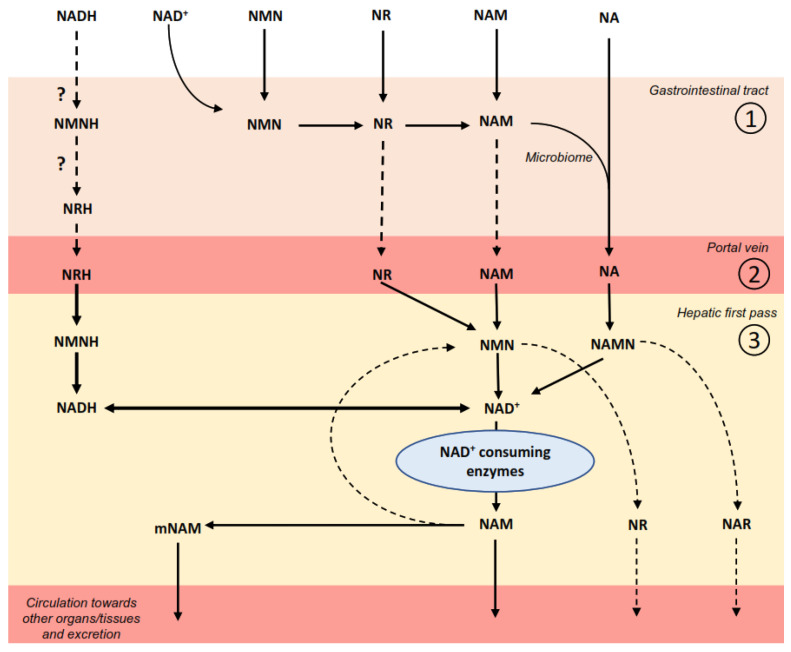
NAD^+^ metabolism after oral administration: The NAD backbone is mostly obtained from the diet via the NAD^+^ and NADH contents of the foods we eat, although other NAD^+^ precursors—such as NAM or NR—can also be found in food items. Moreover, NMN and NA have been used as oral supplements or pharmacological agents. (1) In the gastrointestinal tract, NAD^+^ is sequentially degraded to NMN and NR. NR can be partially absorbed and incorporated in circulation, but a portion of it is further degraded to NAM in the gut lumen. NAM can then be absorbed or further processed to NA by the gut microbiome. (2) NR, NAM, or NA are usually the main molecules reaching the portal vein. Through the portal vein, NR, NAM, or NA can reach the liver (3), where they can be used for NAD^+^ synthesis. NAD^+^ is broken down by multiple families of enzymes, rendering NAM as a product. NAM can then have multiple fates. First, it can be used to regenerate NAD^+^ when transformed to NMN. Additionally, it can be released from the liver to provide NAD^+^ precursors to other organs/tissues. Finally, it can be methylated to methylNAM (mNAM), which is then released, further metabolized, or driven to renal excretion. Intermediates such as NMN or NAMN can be dephosphorylated, generating NR or NA riboside (NAR), which can also be released as precursors for other tissues/organs. The situation for NADH is not well characterized, and the initial steps remain speculative, hence why they are labelled with a question mark in the figure. NADH might be oxidized to NAD^+^ in the gastrointestinal tract, or degraded to NMNH and NRH. The direct oral administration of NRH has demonstrated that NRH can be absorbed as such, detected in circulation, and used for NAD^+^ synthesis in the liver.

**Figure 3 metabolites-12-00630-f003:**
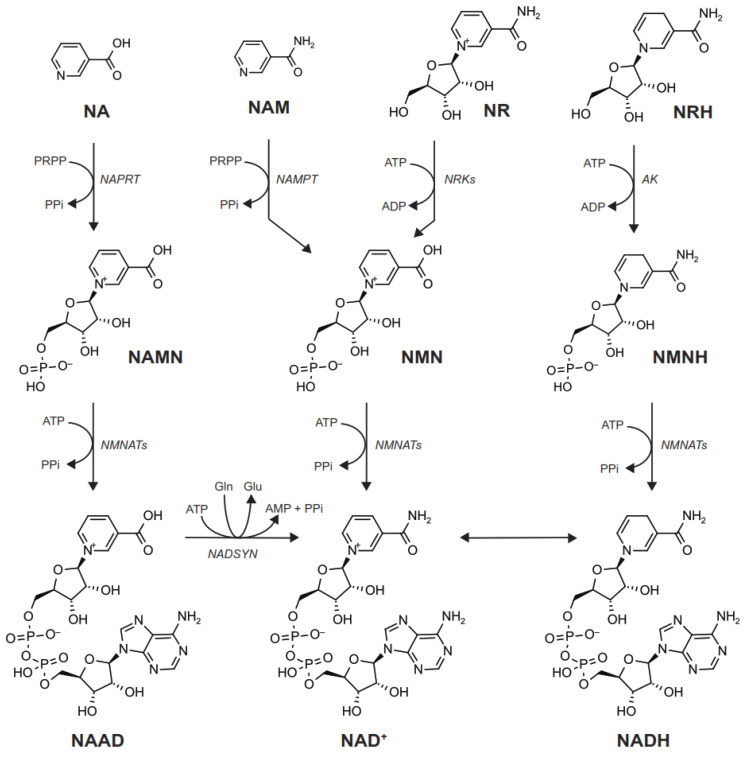
NAD^+^ precursors use different routes and cofactors for NAD^+^ synthesis: The graph illustrates the molecular structure, metabolic reactions, and cofactors required for non-ribosylated (i.e., nicotinic acid (NA), nicotinamide (NAM)) and ribosylated (i.e., nicotinamide riboside (NR), dihydronicotinamide riboside (NRH)) precursors to synthesize NAD^+^. NA and NAM require an initial ribosylation reaction catalyzed by the NA phosphoribosyltransferase (NAPRT) and NAM phosphoribosyltransferase (NAMPT) transferases, using phosphoribosyl pyrophosphate (PRPP) as the ribose donor and releasing inorganic pyrophosphate (PPi). In the case of NA, a second reaction catalyzed by the nicotinamide mononucleotide (NMN) adenylyltransferase enzymes (NMNATs) condenses NA with the adenosine monophosphate (AMP) moiety of ATP, rendering NA adenine dinucleotide (NAAD) and releasing PPi. NAAD is then transformed to NAD^+^ through the NAD synthase (NADSYN) enzyme. In this reaction, NAAD is amidated to NAD^+^, while deamidating glutamine (Gln) into glutamate (Glu). In the case of NAM, its ribosylation leads to NMN synthesis. NMN is then transformed to NAD^+^ via NMNAT enzymes. The two ribosylated precursors, NR and NRH, initiate NAD^+^ synthesis through a phosphorylation step catalyzed by NR kinases (NRKs) and adenosine kinase (AK), respectively. Upon phosphorylation, NR generates NMN, which can be transformed to NAD^+^ by NMNATs. In contrast, NRH generates NMNH, which NMNATs transform to NADH, later being oxidized to NAD^+^.

**Table 1 metabolites-12-00630-t001:** Baseline NAD^+^ levels reported for different cells/tissues. Unless otherwise specified, values refer to murine models.

Tissue/Cell/Matrix	NAD^+^ Levels	References
Liver	300–800 mmol/kg tissue	[11,12,13,14]
Kidney	400–900 mmol/kg tissue	[13,15]
Muscle	300–800 mmol/kg tissue	[11,13,15]
Muscle (human)	300–1000 mmol/kg tissue	[16,17]
Brown adipose tissue	400–700 mmol/kg tissue	[13]
White adipose tissue	20–30 mmol/kg tissue	[11]
Primary hepatocytes	1–4 mol/kg protein	[14]
Plasma (human)	20–80 nM	[18]
Blood (human)	10–20 μM	[17]
PBMCs (human)	10–30 μM	[12]
Urine (human)	30–80 nM	[18]
Cellular nucleus (HEK 293)	87–136 μM	[8]
Mitochondria (HEK 293)	191–275 μM	[8]
Cytoplasm (HEK 293)	92–122 μM	[8]

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
