# Peer review of "NAD+ Precursors: A Questionable Redundancy"

_metabolites, 2022, doi:10.3390/metabo12070630_

Round 1

Reviewer 1 Report

NAD+ is vital in multiple aspects of enzymatic activities and pathways. Overall the manuscript was very well laid out to include multiple levels of NAD+ homeostasis. 

Would it benefit to have a section on NAD+ flux during development or in response to exercise (Grant 2019)?

Author Response

This is a very interesting proposition by the reviewer. However, NAD+ fluxes during development or in response to exercise have not been truly analyzed yet.

Nevertheless I thank the reviewer for pointing out the unintended omission of the interesting manuscript from Grant and colleagues, examining the NAD+ metabolome dynamics after NAD+ intravenous infusion in humans. This sis now discussed in page 5, at the end of section 2.1

Reviewer 2 Report

Presented review is focused on NAD+ precursors. Undoubtedly, the topic of the manuscript is interesting.

During the reading of this article, I had the following major suggestions:

1.       As a result of what kind of transport NAD+ precursors enter the cells ? Are transporters of NAD+ precursors ATP-dependent?

2.       Please include the last chapter presenting clinical trials of NAD+ precursors. Please add also table that summarizes the  effect of presented clinical tials.

Author Response

“As a result of what kind of transport NAD+ precursors enter the cells? Are transporters of NAD+ precursors ATP-dependent?”

The different transporters for NAD+ precursors are not ATP dependent. SLC22A7 and SLC22a13 can act as NA transporters due to their function as anion exchangers (now clarified in page 5, line 205).

NR and NRH are transported through equilibrative nucleoside transporters (ENTs). Hence, by definition, their transport is not directly ATP dependent.

“Please include the last chapter presenting clinical trials of NAD+ precursors. Please add also table that summarizes the effect of presented clinical tials”.

I appreciate this suggestion by the reviewer. However, NAD+ precursors have been used in the clinic for decades. Just for nicotinamide riboside, there have been dozens of clinical trials within the last 5 years. Therefore, I would strongly suggest that the clinical effects of NAD+ precursors should be subject to a dedicated review. Here we simply aimed to evaluate the basic biology of different NAD+ precursors.

If the reviewer and editors consider it appropriate to add a clinical chapter to this review, I would request some more specificity on the expected scope for it.

Reviewer 3 Report

I reviewed the following manuscript entitled: NAD+ precursors: a questionable redundance (Manuscript ID: metabolites-1764855). It is a review paper and this manuscript describes the review about the metabolism of NAD and its related compounds. I feel that the content of this paper is valuable, however, several points should be improved.

 Noticed points

1)      I would like to use more figures and table. For example, the chemical structures of all of the NAD related compounds should be provided. Further, the enzymatic reaction pathways should be presented using figure with their chemical structure and reaction scheme. The use of tables may improve the paper. For example, the relationship between NAD concentration and tissues may a convenient information. These figures and tables should be helpful for readers.

2)      (Minor point) Line 65, Km; The “Km” should be explained.

3)      (Minor point) Figure 2; The relationship between the processes in this figure and the explanation in the figure captions is slightly unclear. The use of (number) or (symbol) may improve this figure.

Author Response

“I would like to use more figures and table. For example, the chemical structures of all of the NAD related compounds should be provided. Further, the enzymatic reaction pathways should be presented using figure with their chemical structure and reaction scheme. The use of tables may improve the paper. For example, the relationship between NAD concentration and tissues may a convenient information. These figures and tables should be helpful for readers.”

Following the suggestion from the referee, I have now added a new figure (Figure 3), depicting the chemical structures of the different NAD+ precursors and their intermediates during NAD+ synthesis. The new figure also includes the cofactors required for the enzymatic reactions.

NAD+ concentrations in different fluids and tissues have been summarized in other recent reviews and articles, as referenced in the text.

“Line 65, Km; The “Km” should be explained.”

Thank you very much for this suggestion. The concept of Km is now explained in page 2, lines 65-71

“Figure 2; The relationship between the processes in this figure and the explanation in the figure captions is slightly unclear. The use of (number) or (symbol) may improve this figure.”

Following the suggestion from the reviewer, I have modified this figure and its legend, by adding a number to some of its parts. I hope this clarifies the flow of the figure.

Reviewer 4 Report

The article is devoted to an actual topic. The review concerns the peculiarities of vitamin metabolism, its involvement in the physiological processes of the body. The article is constructed classically. All relevant aspects were discussed.

Author Response

I thank the reviewer for his/her very kind comments on this manuscript.

Round 2

Reviewer 2 Report

Accept in present form.

Author Response

"Accept in its present form"

I would like to thank the reviewer for his/her time and constructive criticism on this manuscript

Reviewer 3 Report

I checked the revised manuscript entitled: NAD+ precursors: a questionable redundance submitted to Metabolites (metabolites-1764855). Figure 3 has been added in the revised form. However, more information may improve this review paper. For example, tables about the relationship between NAD concentration and tissues is possible?

Author Response

"I checked the revised manuscript entitled: NAD+ precursors: a questionable redundance submitted to Metabolites (metabolites-1764855). Figure 3 has been added in the revised form. However, more information may improve this review paper. For example, tables about the relationship between NAD concentration and tissues is possible?"

Following the reviewer's suggestion, Table 1 has been added to the manuscript. It reports NAD+ levels in a variety of mouse, human and cellular samples